# Current Knowledge on Interactions of Plant Materials Traditionally Used in Skin Diseases in Poland and Ukraine with Human Skin Microbiota

**DOI:** 10.3390/ijms23179644

**Published:** 2022-08-25

**Authors:** Natalia Melnyk, Inna Vlasova, Weronika Skowrońska, Agnieszka Bazylko, Jakub P. Piwowarski, Sebastian Granica

**Affiliations:** 1Microbiota Lab, Department of Pharmacognosy and Molecular Basis of Phytotherapy, Medical University of Warsaw, Banacha 1, 02-097 Warsaw, Poland; 2Department of Pharmacognosy, National University of Pharmacy, 53 Pushkinska Str., 61002 Kharkiv, Ukraine; 3Department of Pharmacognosy and Molecular Basis of Phytotherapy, Medical University of Warsaw, Banacha 1, 02-097 Warsaw, Poland

**Keywords:** dermatology, skin, microbiota, interaction, topical, skin diseases, plant materials, phytotherapy, keratinocytes, fibroblasts

## Abstract

Skin disorders of different etiology, such as dermatitis, atopic dermatitis, eczema, psoriasis, wounds, burns, and others, are widely spread in the population. In severe cases, they require the topical application of drugs, such as antibiotics, steroids, and calcineurin inhibitors. With milder symptoms, which do not require acute pharmacological interventions, medications, dietary supplements, and cosmetic products of plant material origin are gaining greater popularity among professionals and patients. They are applied in various pharmaceutical forms, such as raw infusions, tinctures, creams, and ointments. Although plant-based formulations have been used by humankind since ancient times, it is often unclear what the mechanisms of the observed beneficial effects are. Recent advances in the contribution of the skin microbiota in maintaining skin homeostasis can shed new light on understanding the activity of topically applied plant-based products. Although the influence of various plants on skin-related ailments are well documented in vivo and in vitro, little is known about the interaction with the network of the skin microbial ecosystem. The review aims to summarize the hitherto scientific data on plant-based topical preparations used in Poland and Ukraine and indicate future directions of the studies respecting recent developments in understanding the etiology of skin diseases. The current knowledge on investigations of interactions of plant materials/extracts with skin microbiome was reviewed for the first time.

## 1. Introduction

More and more drugs, dietary supplements, and cosmetic products appear on the pharmaceutical and cosmetic market, which contain medicinal plant materials or substances of plant origin [1,2]. In recent years, phytotherapeutic preparations are gaining greater importance in solving many problems in dermatology and cosmetology [3]. Severe skin ailments require the application of antibiotics, steroids, or calcineurin inhibitors. However, in mild skin disorders, the topical application of plant-based remedies in various pharmaceutical forms, such as raw infusions, creams, ointments, balms, and tinctures can be effective and successfully prevent further disease development [4,5].

Human skin is a complex organ that accounts for about 15% of the total body weight of an adult and has a surface area of 1.5–2 m^2^. This organ is responsible for many vital functions [6]. It protects against external factors; participates in thermoregulation, metabolism, the regulation of fluid balance, and body shape maintenance; and eliminates toxins from the body by sweat excretion [7]. It consists of several layers, such as the living tissue of the dermis, epidermis, and the outer-facing layer, which is called the stratum corneum [8].

On the microanatomic level, skin is composed of keratinocytes, Langerhans cells, fibroblasts, mast cells, macrophages, endothelial cells, and lymphocytes, which form a complicated and fine-tuned organization such as the skin immune system [9,10]. Likewise, the superficial layer of the skin is home to millions of bacteria, fungi, and viruses that compose the skin microbiota [8]. This microbial ecosystem supports many skin functions, including metabolism, vitamin synthesis, protection against pathogen invasion, immunity development, and regulation [11].

Currently, an increase in the incidence of skin diseases worldwide is observed, which is mostly linked to the adverse factors of modern civilization [12]. For instance, 42 of the 145 surveyed people in Poland reported past or present skin disorders [13]. In Ukraine, the morbidity rate of skin diseases increases every year. The currently reported problems mostly relate to dermatitis, atopic dermatitis, eczema, psoriasis, wounds, and others. Many factors influence skin well-being, with the highest contribution attributed to domestic detergents, cosmetics, and antiseptics.

The present review aims to summarize the current information on the plant materials that are contained in pharmaceutical and cosmetic preparations available in Poland and Ukraine and marketed as non-prescription medicines or medical products. The paper presents the current knowledge regarding the traditionally used plant raw materials related to their influence on the skin and skin microbiota. Medicinal plants used in treating wounds, burns, dermatitis, atopic dermatitis, eczema, and other skin inflammatory diseases have been included.

Based on the scientific database Scopus, a graphical representation of the publication trend in the field of the present review was generated. The trend in the number of papers focused on plant extracts or materials plants and skin shows a significant increase over the last 20 years. Around 190 reports were published in 2001 according to Scopus, and in 2021, it was over 1300 original papers and reviews. Similarly, a dynamic increase in the interest in research devoted to plant preparations and microbiota was observed (Figure 1) in 2001 when 4 papers matching chosen keywords were recorded compared to 377 in 2021. This analysis confirms that the general interest in the different aspects of the interaction of plant materials with skin increases, and the subject is worth further investigations. This trend can be explained by the fact that natural therapies are gaining much interest in the context of changes in the lifestyle of the population around the world.

## 2. Methods Used in the Review

A literature search was conducted using scientific databases such as PubMed, Scopus, and Wiley Online Library for relevant studies with the keywords “microbiome”, “microbiota”, “commensal”, “topical”, “plant materials”, and “phytotherapy”, “dermatology”, “skin”, “skin disorders”, “keratinocytes”, “fibroblasts”. All search terms were used in various combinations, and studies were screened for relevance based on their abstracts. Studies written in English, Polish, and Ukrainian languages were considered.

The selection of non-prescription medicines and medical products was based on the search of topically used remedies containing material of plant origin in the Ukrainian directory of drugs “Compendium” and on the Polish pharmacy websites. Moreover, the Ukrainian online service “Tabletki.ua”, which provides information on the availability of medical preparations and other pharmaceutical products in pharmacies, was considered.

## 3. Human Microbiota in the Skin Inflammation Process

The skin is the habitat and a source of nutrients for various symbiotic, commensal, and pathogenic microorganisms described as skin microbiota [7]. Early studies showed that abundant bacterial genera on the surface layers of the human skin include *St**aphylococcus*, *Propionibacterium*, *Micrococcus*, and *Corynebacterium* [14]. The composition of the human microbiome similarly varies quantitatively and qualitatively according to the body site on which it is located, depending on the distinctive characteristics (pH, moisture, salinity, and sebum content), and may also vary due to other factors (e.g., genotype, age, and sex) [5,12]. The recent surveys have a greatly advanced understanding of the host-symbiont and host-pathogen relationships and established that the skin microbiota plays a beneficial role, much like the gut microbiota, indicating that the bacteria present on our skin have similar functions in immune regulation and disease pathogenesis [11,13,14]. 

Commensal bacteria can passively occupy a similar ecological niche to a pathogenic microbe, thus impeding its skin colonization. Additionally, commensal bacteria on human skin can selectively induce antimicrobial peptides production and provide a protective effect in vivo when administered before the infectious challenge. The human microbiome may also modulate the immune system, directing it to eliminate the disease-causing factor [14,15].

## 4. Interaction of Plant Origin Material with Microbiota

The interaction of microorganisms with plants can be considered from two sides. On the one hand, there is the influence of plants on microbes’ growth and metabolic functions, as medicinal plants contain extractable biochemical and bioactive compounds, which can target certain viruses, bacteria, or fungi [16]. On the other hand, a metabolically active microbial community can alter the chemical structure and composition of natural products applied to the skin.

To date, there are a lot of known natural products which act effectively against bacteria, fungi, viruses, or protozoa. For instance, phenolics have antifungal and antiviral properties, and it is established that their overall harmfulness to microorganisms is associated with the measure of hydroxyl bunches and their locations on the phenol bunch. Low dosages of phenols (0.032%, 320 g/mL) destroyed fast-developing cultures of *Staphylococci* and *Streptococci* [17]. Quinones have been found to frame irreversible edifices with nucleophilic amino acids in proteins, leading to protein inactivation and function loss resulting in antibacterial effects. Quinones may also make it difficult for bacteria to obtain substrates [18]. The antimicrobial mode of action of tannins is linked to their ability to inactivate microbial adhesins, enzymes, and cell envelope transport proteins [19].

Several in vivo and in vitro examinations demonstrated that different plant preparations repress bacterial species found in cutaneous diseases. Chamomile essential oil and bisabolol were found to have activity mostly against Gram-positive bacteria, *Staphylococcus aureus*, *Bacillus subtilis*, and the fungus *Candida albicans*. Aqueous extracts from *Allium cepa* showed antifungal activity against *Malassezia furfur*, *Candida albicans*, other *Candida* sp., and other dermatophyte species [17].

In terms of the influence of microbiota on plant materials, it should be mentioned that the literature review showed a lack of information about the skin microbiota metabolism. However, nowadays, many investigations on microbiota residing in the gut have been described. For instance, human and swine microbiota transformed natural products in the goldenrod infusion into smaller molecules, mainly phenylpropanoid acid derivatives [20].Twenty metabolites were detected and characterized after incubating the linden flower extract with human gut microbiota [21]. All changes in the chemical composition of the raw plant material caused by the interaction with microbiota can lead to potentially active compounds responsible for their bioactivity in vivo. This is also evidenced by experiments on the gut microbiota-derived metabolites of ellagitannins-urolithins. It was established that urolithins inhibit proinflammatory cytokines expression in RAW 264.7 macrophages, which has an important role in inflammatory bowel diseases [22].

## 5. Phytotherapy in Skin Diseases

Patients and physicians widely use phytotherapy throughout the whole world. This type of therapy is as old as humankind. Plant-derived drugs are in demand because of several advantages, such as often having fewer side effects and better patient tolerance. Apart from this, treatment with natural products is more affordable to patients than chemical medicines.

Clinical trials and in vitro and in vivo experiments were conducted for many plant materials, showing their effectiveness in inhibiting the formation of cytokines and eicosanoids, preventing the inflammatory reaction cascade. However, still, the use of most herbal medicines is based solely on their longstanding traditional use in folk medicine [23].

Over the years, plants that deserve special attention in treating skin diseases have been identified. Below is a list of plant materials that are the most popular in the traditional treatment of skin diseases in Poland and Ukraine (Table 1).

Different herbal parts are used in treating skin ailments, but the most usable are the above-ground parts, such as herbs, leaves, or flowers. Among all plant effects are frequently noted anti-inflammatory, bactericidal, and wound healing effects. The mostly described plants are traditionally used for minor superficial wounds, burns, minor inflammations of different etiology, irritations, bedsores, and ulcers. However, for example, *Chelidonii herba* is applied to warts, corns, and pimples due to their antiseptic and fungicidal effects, and *Melissae folium*, like antiviral, antimicrobial external remedies for herpes.

Polish and Ukrainian pharmaceutical markets have been analyzed and preparations marketed as non-prescription medicines or medical products containing plant material from Table 1 as an active ingredient are presented in Table 2.

In Poland, as in Ukraine, the most popular plant material for treating skin diseases is marigold flowers (*Calendulae flos*). It is in pharmaceutical forms, such as ointments or tinctures, and is produced by different manufacturers. It is used externally in treating mild inflammatory conditions of the skin and ancillary in treating minor skin injuries. At Polish pharmacies, a large number of remedies from linseed, chamomile, and tormentil are found. Tormentil is presented mostly in ointments (“Tormentiol”, “Tormentile Forte”, and “Tormentillae unguentum compositum”), linseed in ointments and creams (“Linomag”), and chamomile in ointments, gels, and tinctures (“Kamagel” and “Azulan”). Ukrainian pharmacies have some distinguishing topical remedies of *Hippophae rhamnoides*, such as “Sea buckthorn ointment”, “Olasol spray”, and some balms.

The plant materials were selected for the literature surveys based on the above approach. The plants which are considered effective in the treatment of inflammatory skin diseases are discussed below. Additionally, to information on the chemical composition, activity studies, and traditional use, data were also searched for in studies of the effect of skin microbiota on natural products contained in the listed plant materials and/or the analyses of changes in the composition of the microbiome as a result of the use of preparations from the raw materials mentioned above.

### 5.1. Allium cepa L. (Onion Bulbs)

The plant substance of *Allii cepae bulbus* consists of thick and fleshy leaf sheaths and leaf approaches from *Allium cepa* L. (Amaryllidaceae). Biologically active compounds of onion bulbs are flavonoids and sulfur-containing compounds. In traditional medicine, onions have been used externally to treat insect bites, wounds, minor burns, boils, warts, and bruises [27]. Many studies have demonstrated the antioxidant and antimicrobial activity of the extracts [28]. The study using human fibroblasts showed that the onion extract inhibited their proliferation, induced apoptosis, and decreased expression of β1 integrin, which may be beneficial in treating keloid and hypertrophic scars [29]. However, data on the activity of onion extracts in treating hypertrophic scars are inconclusive. 

The effect of Mederma (Merz Pharma, Frankfurt, Germany) gel, containing *Allium cepa* as the active ingredient, was investigated in a rabbit hypertrophic scar model. No significant effect of the gel on the reduction in scar hypertrophy, vascularization, or inflammation was determined, but an improvement in the organization of dermal collagen was observed [30]. The effectiveness of the same gel was investigated in patients with new surgical scars compared to petrolatum emollient. No significant differences were found in the activity of these formulations in the treatment of scars as assessed by redness, itching, burning, pain, thickness, or overall cosmetic appearance [31]. Another study comparing the activity of Mederna with a petroleum-based emollient assessed the appearance and symptomatology of postoperative scars. The onion extract gel was ineffective, no difference was found in the evaluation of the redness and itching of the scar after one month of use. In contrast, a reduction in redness has been observed in patients using the emollient [32].

The effectiveness of this plant material in treating scars after minor dermatological procedures has been proven for the occlusive overnight intensive patch medical device containing onion bulbs extract and allantoin. After 24 weeks, a clear improvement in the appearance of the scar was observed, assessed with the Patient and Observer Scar Assessment Scale and a Global Aesthetic Improvement Scale [33]. The beneficial effect in the treatment of scars has also been confirmed for other combined preparations containing *Allium cepa* extract. The gel enriched with allantoin and heparin (Contractubex, Merz Pharma, Frankfurt, Germany) improved vascularization, pigmentation, and the overall appearance of the scar according to the Vancouver Scar Scale [34]. The use of a gel containing *Allium cepa* extract, allantoin, and pentaglycan (Kaloidon gel, Laboratorio Farmacologico Milanese SRL, Caronno Pertusella, Italy) for 24 weeks successfully reduced neoangiogenesis in patients with hypertrophic scars and keloids, resulting in the clinical improvement of skin lesions [35]. Moreover, using a patch containing 10% *Allium cepa* extract, 1% allantoin, and 4% pentaglycan (Kaloidon patch, Laboratorio Farmacologico Milanese SRL, Caronno Pertusella, Italy), after 24 weeks, showed beneficial effects according to the Patient and Observer Scar Assessment Scale. In addition, significantly improved skin scar thickness and vascularization were observed after 12 weeks [36].

There are two medicinal products available on the Polish market, containing *Allium cepa* extract, allantoin, and heparin. It is a Contractubex (Merz Pharma, Frankfurt, Germany) gel and Cepan (Unia, Warsaw, Poland) cream, which additionally contains chamomile extract. In the literature, no reports on the interactions of extracts from onion bulbs with skin microbiota were found.

### 5.2. Aloe vera L. (Aloes Leaves)

*Aloe vera* is a plant that belongs to the Asphodelaceae family. *Aloe vera* leaf gel is mainly used in dermal ailments. This gel is rich in polysaccharides. Glucomannan, acetylated glucomannan, galactogalacturan, glucogalactomannan, and acemannan had been extracted and described from this *Aloe* species [37]. Moreover, such sterols as lupeol, campesterol, and β-sitosterol were also found [37]. 

In the past, the influence on skin cells was investigated. In vitro pharmacological studies determined that the effect of *Aloe vera* gel and its compounds on HaCaT cells lies in decreasing photodamage; maintaining membrane integrity; reducing the levels of TNF-α, IL-8, IL-12 and p65; and increasing IκB-α protein expression. *Aloe vera* gel, in turn, increases the wound healing ability, number of cells, keratinocyte proliferation and differentiation, and cell surface expression of adhesion molecules (β1-integrin, α6-integrin, β4-integrin, and E-cadherin) in HEKa [38]. In vivo pharmacological studies on *Aloe vera* were also conducted and showed an increasing amount of fibroblasts, TGF-β gene expression, wound closure and skin tensile strength, collagen deposition, and wound healing activity as re-epithelialization and angiogenesis [38]. Moreover, it was established that *Aloe* sterols reduce skin dryness, epidermal thickness, wrinkle formation, and pro-inflammatory cytokines levels, and lupeol, campesterol, and β-sitosterol are significantly anti-inflammatory in wounded mice [37]. Polysaccharides isolated from *A. vera* help to regulate the wound healing activity, inducing matrix metallopeptidase (MMP)-3 and metallopeptidase inhibitor-2 gene expression during the skin wound repair in rats [39].

In most cases, aloes occur in complex medical products as additional ingredients. On the Ukrainian market, it is presented in creams “Express BITE” and “Express Burn” (Georg BioSystems, Kirovograd, Ukraine).

### 5.3. Arnica montana L. (Arnica Flowers)

*Arnica montana* is a widely used therapeutic plant belonging to the Asteraceae family. This plant possesses numerous medicinal activities due to such constituents as flavonoids, sesquiterpene lactones (metacryl, isobutyryl, tygloyl, methacryloyl, and isovaleryl helenalin derivatives), acetylenes, hydroxycoumarines (umbelliferone and scopoletin), phenyl acrylic acids, essential oil components, and phenolic acids (chlorogenic and caffeic acid). It is also known to contain pyrrolizidine alkaloids (tussilagin and isotussilagin) [40]. It has been used for centuries in dermatology as an antiphlogistic, antibiotic, and anti-inflammatory remedy [41]. The pharmaceutical form in traditional topical use is presented as herbal preparations in semi-solid and liquid dosage forms for cutaneous use [42].

Lyss et al. investigation shows that the main anti-inflammatory sesquiterpene lactone from arnica, helenalin, modifies the NF-κΒ/ΙκΒ complex, preventing the release of ΙκΒ [38,40]. Some results present that arnica reduced the UVB-induced inflammatory response as demonstrated by the inhibition of myeloperoxidase activation, a decrease in NF-κB levels, and a reduction in proinflammatory cytokines levels (IL-1β, IL-6, TNF-α, and IFN-γ) in in vivo studies [43]. In some in vitro experimental models, the production of IL-6, IL-8, and TNF-α pro-inflammatory cytokines was also measured. The secretion of IL-6, IL-8, and TNF-α in an H_2_O_2_-stressed fibroblast cell culture decreased, which indicates the cytoprotective effect against cell membrane oxidative damage and higher anti-inflammatory activity [42,44].

Arnica is a popular and characteristic plant material for the Polish market and two OTC medicines with extracts from this plant material are available at Polish pharmacies. “Arnithei” gel (Dr. Theiss Naturwaren, Homburg, Germany) contains an arnica tincture and is used for relieving bruises, sprains, and local muscle pain. “Uzarin” gel (Nes Pharma, Tarnów, Poland) contains the extract of *Arnica montana* and *Calendula officinalis* and is applied for bruises, swellings, first-degree burns, and after insect bites. In some sources, the beneficial effect of arnica extracts on the composition of skin microbiome is mentioned; however, there are no fine basic studies supporting these statements [45].

### 5.4. Calendula officinalis L. (Marigold Flowers)

*Calendula officinalis* (Asteraceae) is one of the most popular plants used clinically throughout the world [46]. This plant contains triterpene saponins (2–10%), mainly oleanolic acid glycosides; free and esterified triterpene alcohols, especially faradiol 3-mono- and diesters; carotenoids (up to 3%): α- and β-carotene, lutein, and rubixanthin; flavonoids (0.3–0.8%) based on quercetin, quercitrin and isorhamnetin; polysaccharides; sterols; sesquiterpenoids (aloaromadendrol and epicubebol); phenolcarbonic acids; fatty and amino acids; tocopherols; and essential oil (0.2–0.3%) with α-cadinol and β-cadinen as the major components [24,25,47,48]. Frequently, the infusion, tincture, and ointment of marigold are used as a wound healing remedy for inflammation of the skin and mucous membranes and externally in the treatment of long-healing wounds, cuts, boils, burns, and ulcers [25,48,49].

Pharmacological studies had confirmed that extracts from this plant exhibit a broad range of biological effects, such as antibacterial, antifungal [47,48,50,51], antioxidant [47,52], anti-inflammatory [47,48,51,52], spasmolytic, anticancer [24,53], anti-HIV, and hepatoprotective activities [51], and stimulate the proliferation and migration of fibroblasts in vitro [54]. In fact, most of the studies on marigold focus on its anti-inflammatory property. Some results present that *C. officinalis*, with other plants, reduced cutaneous inflammation at the price of the downregulation of inflammatory IL-1β, IL-6, and IL-8 and suppressed an increase in stratum corneum dehydration through the upregulation of AQP3 [55]. Marigold flowers extracts protect HaCaT skin cells against an oxidative stress challenge in the form of H_2_O_2_ [56]. Moreover, other scientists found that the *n*-hexane and the ethanolic extracts modulated the inflammatory phase of wound healing by activating the transcription factor NF-κB and increasing the amount of the chemokine IL-8 [57].

Nowadays, marigold flowers are one of the most famous plant materials among manufacturers and patients as they are presented in large amounts in Polish and Ukrainian pharmacies. For patients, it is available mainly in two forms (ointments and tincture) and is used in mild inflammations of the skin, minor wounds, burns, and cuts. One study on the potential influence of the marigold extract on the composition of skin microbiota was performed. The 90% hydroethanolic extract from marigold was shown to inhibit the growth of *P. acens* and *S. epidermidis*. The results suggest that this plant material can be considered a skin prebiotic important in treating acne [58].

### 5.5. Chelidonium majus L. (Greater Celandine Herb)

*Chelidonium majus* L. is also known as greater celandine (family Papaveraceae). The herb of *Chelidonium majus* L. contains over 20 different alkaloids, including chelerythrine, chelidonine, sanguinarine, isochelidonine, and protoberberines (berberine, coptisine, dihydrocoptisine, and stylopine) protopine [59]. Several flavonoids were found in the aerial parts in low amounts. Among them are derivatives of kaempferol and quercetin. Moreover, other phenolic compounds such as hydroxycinnamic acids, hydroxybenzoic acids, and their derivatives were identified. Additionally, organic acids (chelidonic, malic, citric, and succinic acids), biogenic amines (histamine, methylamine, and tyramine), essential oil constituents, triterpenoids, saponins, vitamins A and C, and nicotinic acid were found in *C. majus* extracts [60].

N. Cordes et al.’s investigation shows that ukrain, an alkaloid thiophosphoric acid derivative of *C. majus*., demonstrates a protective effect in normal human fibroblasts in modulating radiation toxicity [61]. Vavrecková et al. determined the antiproliferative activity of the extract on human keratinocytes, showing IC_50_ was lowest for sanguinarine (2.26 µM), extract (as chelidonine) ca. 5.68 µM, chelidonine, and chelerythrine ca. 28 µM, and poor activity of berberine and hydrastinine. The lactate dehydrogenase assay showed the cytostatic activity of the *C. majus* extract rather than cytotoxic activity which can be considered beneficial in treating wards [62]. Some medical products with celandine extracts are available on the Ukrainian market and presented in such preparations as “CHYSTOTIL” cream (Khimpharmzavod Chervona Zirka, Ukraine) and “CHYSTOTIL” ointment (NATURE LIFE, Ukraine). Bactericidal, wound healing, anti-inflammatory, and analgesic properties are indicated.

### 5.6. Hamamelis virginiana L. (Witch Hazel Leaves, Bark)

*Hamamelis virginiana* L., also known as witch hazel, is a shrub that belongs to the Hamamelidaceae family [63,64]. Preparations from *Hamamelis* leaves, bark, and twigs, present in extracts, tinctures, creams, and salves, are utilized to treat dermatological (sunburn, irritated skin, and atopic eczema) and vascular disorders (hemorrhoids, varicose veins, and phlebitis), highlighting the fact that this plant has a wide range of biologically active substances [64,65,66].

Witch hazel bark contains up to 10% tannins (hamamelitannin and catechins), free gallic acid, and a small amount of flavonols, fats, and waxes. Leaves contain 3–10% of tannins (a mixture of gallotannins and condensed catechins–procyanidins); notably a small amount of hamamelitannin; phenolic acids (caffeic and gallic acids); flavonoids such as kaempferol, quercetin, quercitrin, and isoquercitrin; and essential oil [26,64,67,68,69]. Numerous in vitro and in vivo studies have shown that this plant has antitumoral, antioxidant, anti-inflammatory, antibacterial, antiviral, and antimutagenic activity [63,64,68,69,70]. For instance, the extract of witch hazel leaves and small twigs can decrease the amount of IL-8 produced by fibroblast cells [71]. The hazel extract and its component—hexagallloylglucose—regulated the inflammatory response via inhibiting NF-κB and PAR-2 pathways in human keratinocytes [72].

Witch hazel, one of the active ingredients, was found in the combined homeopathic ointment TRAUMEEL^®^ S (Biologische Heilmittel Heel GmbH, Baden-Baden, Germany), which is available on the Ukrainian market and applied for the treatment of bedsores, burns, pityriasis, and trophic ulcers. No papers reporting the possible interactions of hazel extracts with human sin microbiome were found.

### 5.7. Hippophae rhamnoides L. (Sea-Buckthorn Fruits)

*Hippophae rhamnoides* L., (Elaeagnaceae family), is commonly known as sea buckthorn. Fruits and seeds contain fatty oil (about 8% in fruits and about 12% in seeds), which contains a significant amount of carotene (up to 250 mg%), vitamins E, F, and K, phospholipids (up to 1%), and fatty acids (linoleic, oleic, palmitic, palmitoleic, and stearic). Fruits contain mono- and disaccharides, mucus, vitamins (C, B_1_, B_2_, B_6_, B_9_, P, and PP), organic acids (malic, tartaric, oxalic, and succinic), sulfur-containing substances, including betaine and choline, tannins, flavonoids (rutin, quercetin, kaempferol, and isorhamnetin), phenolic acids (chlorogenic and caffeic), and coumarin [25]. Patients have used sea buckthorn for a long time due to its rich composition, which provides a wound healing effect, modification of sebum characteristics, and improvement of atopic skin [73].

To date, many investigations have been conducted on seeds and their non-polar compounds. It was shown that the palmitic acid-enriched fraction supported cell proliferation properties on normal human keratinocytes (NHEK) and normal human dermal fibroblasts (HDFa). However, some fractions did not alter the cellular morphology of normal keratinocytes and did not influence the inflammatory response [73]. There are also studies confirming that sea buckthorn seed oil stimulated the proliferation of dysplastic cells, while it also impaired the ability of both normal and dysplastic cells to migrate over a denuded area [74]. 1,5-dimethyl citrate isolated from *Hippophae rhamnoides* was demonstrated to prevent LPS-induced NO production and inhibited the expression of IKK-α/β, IκB-α, NF-κB p65, iNOS, and COX-2 and the activities of IL-6 and TNF-α [75].

Sea buckthorn ointment (Fitolic, Ivano-Frankivsk, Ukraine) and Olasol spray (STOMA, Kharkiv, Ukraine) are drugs popular in Ukraine and contain sea buckthorn oil, used for healing infected wounds, including long-term non-healing burns, trophic ulcers, and skin grafts. Moreover, some other medical products consisting of *Hippophae rhamnoides* extract are available, such as the balms Mintalon (Minta, Kharkiv, Ukraine) and Reskinol (Botany, Kramatorsk, Ukraine). To the best of our knowledge, the interaction of sea buckthorn extracts with skin microbiota has never been investigated.

### 5.8. Linum usitatissimum L. (Linseed)

Linseed is usually defined as the ripe dried seeds of *Linum usitatissimum* L. (Linaceae), which contain 30–45% of fixed oil, 25% protein, 3–9% polysaccharides, and 0.1–1.5% cyanogenic glycosides. In addition, it also contains lignans, mainly secoisolariciresinol, and its glycosides [76]. It is a herbal medicinal product with well-established use in treating habitual constipation [77]. In traditional medicine, flax seeds have been used to relieve inflammation of the upper respiratory tract and gastrointestinal tract and externally in skin inflammation, eczema, ulcers, burns, chilblains, hard-to-heal wounds, skin drying, and cracking [78]. 

On the Polish market, some drugs contain virgin linseed oil, indicated in the treatment of skin diseases such as eczema and rash, as well as in conditions of excessive dryness and symptoms of psoriasis. In addition, there are preparations containing flax in medical products that intensely moisturize, soften, and nourish the skin during or after radiotherapy.

In vivo studies have shown that the external application of linseed oil has an anti-inflammatory effect. In rats with carrageenan-induced paw edema, a reduction in clinical signs of inflammation, infiltration of inflammatory cells, vascular congestion, and an improvement in biochemical parameters were observed [79].

The effectiveness of treating burns with linseed oil has been proven in animal models. After applying linseed oil to second-degree burns in rats, a higher wound closure rate was observed, and the performed biopsy showed better tissue regenerative properties and higher angiogenesis than the control [80]. In another study, also on the model of second-degree burns in rats, it was confirmed that after 21 days of linseed oil application, the severity of the inflammatory process decreased, and collagen synthesis, re-epithelialization, and angiogenesis increased [81]. In a burn model in rabbits, after 12 days of treatment with linseed oil, the degree of wound closure was significantly higher than in the control group; moreover, complete wound closure was 9 days earlier than in the control group. A histopathological study showed that in the group treated with linseed oil, the wound contained fewer inflammatory cells and had complete re-epithelialization with reduced thickness compared to the non-treated control. In addition, an increased number of new capillaries, collagen fibers, and fibroblasts was observed [82]. The effectiveness in treating burns was also confirmed for a gel containing flax seed polysaccharides (composed of glucose, mannose, xylose, and arabinose in glycerol) in the rats’ burns model. The group treated with the gel showed the best results. The skin was naturally colored, and a histological evaluation showed epidermal regeneration without inflammation, growth in connective tissue, and increased collagen production [83].

The beneficial effects of flax have also been confirmed in wound healing. In wound models made in rats, it was shown that the application of linseed oil accelerates wound closure, increases re-epithelialization, and reduces inflammation [84,85]. The histopathological examination also showed the increased synthesis of collagen fibers, vascularization, and hair follicles [86]. Linseed oil used on wounds made with a scalpel in rabbits increased the skin’s elasticity and firmness and stimulated microcirculation and the influx of fibroblasts, as well as the growth of collagen fibers [87]. No information on the research focusing on interactions of flax extracts with skin microbiota was found in the literature.

### 5.9. Matricaria chamomilla L. (Chamomile Flowers)

Flowers from *Matricaria chamomilla* L. (Asteraceae) contain not less than 4 mL/kg of essential oil and 0.25% of apigenin-7-glucoside. In addition to essential oil and flavonoids, they contain coumarins, phenolic acids, and polysaccharides [88]. The indications for external use include the treatment of minor ulcers and inflammation of the mouth and throat, inflammation of the skin (sunburn), superficial wounds and small boils (furuncles), and as an adjunct to the treatment of skin and mucosa irritation around the anus and genital region [89].

The anti-inflammatory properties of the essential oil and water extract of chamomile flowers have been confirmed in animal models. In a model of rats with carrageenan-induced paw edema, they reduced swelling and decreased prostaglandin E_2_ secretion and NO levels. In a mouse model with xylene-induced ear swelling, they reduced the swelling and lowered the allergic reaction. Additionally, the essential oil reduced the duration and frequency of scratching in mice with dextran-induced itching [90].

The acceleration of the second-degree burn regeneration in rats was observed for the chamomile flower oil extract, which significantly reduced the lesion area after 20 days [91]. In addition, another study showed that the same extract reduced the size of the incision made on the back of rats after just 5 days. Complete healing was observed after 11 days, while for olive oil alone, the healing process took 20 days [92].

The ability to regenerate wounds was also confirmed for ethanolic and methanolic extracts from *M. chamomilla* flowers. The daily use of the gel containing 5 or 10% ethanolic extract improved wound healing in diabetic rats by increasing fibroblast proliferation and revascularization. A higher wound closure ratio was noted after three days of treatment compared to the control group. No significant differences were observed between the treatment with the gel containing 5 and 10% of the extract [93]. The wound healing capacity of the methanolic extracts was tested in an excision wound model on the rats’ dorsum, using concentrations of 2.5, 5, and 10%. Seven days after the injury, a difference in wound regeneration was observed between the test and control groups, the ointment containing 10% of the extract had the strongest healing effect. After 11 days, the differences in wound healing after the application of the 2.5, 5, and 10% ointments were insignificant, and still, all of them aided the healing more strongly than the control. Histopathological examinations carried out after 14 days showed that using the ointments with the chamomile flower extract increased the number of fibroblasts, basal epidermal cells, and the amount of collagen. In contrast, the number of neutrophils in the wound decreased [94]. The activity of the ethanolic extract from chamomile flowers was also tested in a model of wounds infected with *Staphylococcus aureus* in mice. After 14 days of daily use, the wound was completely healed, and the hair was present, without scarring, while scar tissue appeared in the gentamycin-treated group and severe inflammation in the control group. Moreover, in the group treated with chamomile extract, an increase in granulation tissue production, fibroblast density, keratinization on the wound surface, and thickness of collagen fibers was observed [95].

The clinical efficacy of the *M. chamomilla* flower extract applied to the forearm and face of healthy volunteers was tested. Skin physiology was assessed after 2 h and 2 and 4 weeks of daily use. The use of the extract significantly increased the hydration of the corneum, and after prolonged use, it reduced the transepidermal water loss by 27% [96]. Another study looked at the activity of chamomile gel in preventing acute radiation dermatitis in head and neck cancer patients. The effect of a gel containing 8.35% of chamomile flower extract was compared with urea cream. The use of the gel delayed the onset of the inflammatory reaction to the radiation. In addition, its use was associated with a lesser incidence of itching, burning, and discoloration among patients, which were seen in the group using urea cream [97].

The activity of 3% of *M. chamomilla* essential oil was tested in a BALB/c mouse model in which atopic dermatitis was induced with dinitrochlorobenzene. The daily use of the oil for four weeks contributed to a decrease in the levels of Ig E and Ig G1 and histamine in the blood of the animals. In addition, it reduced the frequency of scratching [98]. Although several basic and clinical studies were performed with chamomile extract as a skin medicine, no research considering its influence on skin microbiota has been reported so far.

### 5.10. Potentilla erecta L. (Tormentil Root)

*Tormentillae rhizoma* is a whole or cut dried rhizome of *Potentilla erecta* (syn. *Potentilla tormentilla*, Rosaceae), containing not less than 7% of tannins expressed as pyrogallol. The composition is dominated by condensed tannins (up to 22%), but ellagitannins (including agrimoniin and pedunculagin) are also present. In addition, there are phenolic acids (coumaric, sinapic, caffeic, and gallic acids and their derivatives), flavonoids (kaempferol and quercetin and their derivatives), as well as triterpene saponins [99].

The indications for use exclusively based on longstanding use only include the symptomatic treatment of mild diarrhea and mild inflammation of the oral mucosa [100].

On the Polish market, ointments containing a liquid extract from the rhizome of common tormentil are available. It is used together with zinc oxide and ichthammol. The indication for the use of these preparations is the treatment of minor skin lesions, such as scratches or abrasions of the epidermis. In the case of ointments that additionally contain borax, the indications for use are extended to the treatment of purulent and acne lesions. 

Studies on in vitro models have shown the astringent, antimicrobial, antioxidant, and anti-inflammatory effects of tormentil rhizomes [101].

The potent anti-inflammatory properties of the agrimoniin-rich fraction were confirmed in in vitro and in vivo models of UVB-induced inflammation. The investigation using HaCaT keratinocyte cells showed that the abovementioned fraction reduced the production of prostaglandin PGE_2_ by inhibiting COX-2 [102]. In similar studies on the HaCaT cell model, it was demonstrated that the tormentil ethanolic extract inhibits the activation of NF-κB and, in addition to inhibiting PGE_2_ production, also inhibits the production of IL-6 [103]. In the in vivo model of erythema induction on the skin of healthy volunteers, a significant reduction in inflammation and redness was observed after the use of the agrimoniin-rich fraction in the concentration of 100 mg/mL [102]. The effect of the methanolic extract from *P. erecta* rhizomes on the healing of diabetic wounds was investigated in Wistar rats with streptozocin-induced diabetes. Studies have shown that the extract significantly accelerates wound contraction compared to control and increases nitric oxide, glutathione, and collagen levels, while the thiobarbituric-acid reactive substances levels decreased [104]. No studies on the influence of the tormentil rhizome extract on skin microbiota or the metabolism of natural products contained in this plant material by skin microorganisms have been reported.

### 5.11. Quercus robur L. (Common Oak Bark)

Common oak belongs to the family Fagaceae and contains a highly variable amount of tannins (8–20%). *Quercus cortex* contains hydrolyzable tannins (gallotannins, ellagitannins, and flavonols-ellagitannins) and condensed tannins (proanthocyanidins). More than 20 compounds (catechins and low-molecular-mass, oligomeric, and polymeric proanthocyanidins) have been isolated from the bark [105,106]. Triterpenes, insoluble lipid polyesters, and volatile acids are also presented in oak bark. It is considered a traditional herbal medicinal product for the symptomatic treatment of minor inflammation of the oral mucosa or skin in pharmaceutical forms, such as infusion and decoction [105,106].

Ji-Ae Hong et al. found that *Quercus fruits* rescued UVB-induced cytotoxicity and substantially inhibited cellular ROS production in human keratinocytes [107]. Likewise, this study showed that *Quercus fruits* effectively prevent skin photoaging by enhancing collagen deposition and inhibiting MMP-1 via the ERK/AP-1 signaling pathway. *Quercus mongolica* and its isolated compounds have shown inhibitory activities toward inflammatory cytokines and chemokines. Potent activities against MCP-1, TARC, IL-6, IL-8, IL-10, and IL-13 in keratinocytes irradiated with UVB were determined [108]. Chang Seok Lee et al.’s study revealed that oak wood vinegar has anti-inflammatory and antiproliferative effects in a 2,4-dinitrochlorobenzene-induced contact dermatitis mice model. Furthermore, they showed that the mechanism by which oak wood vinegar most likely inhibits epithelial proliferation is through STAT3 inactivation [109].

In most cases, oak bark is used in decoctions by humankind, but in the Ukrainian market, this plant material was also found in a combined medical product as cream, “Bioflorin” (Khimpharmzavod Chervona Zirka, Kharkiv, Uktaine), which has anti-inflammatory and wound healing properties. No information on the interactions of oak bark extracts with sin microbiota was found in the available literature.

### 5.12. Salvia officinalis L. (Sage Leaf)

*Salviae folium*, obtained from *Salvia officinalis* L. (Lamiaceae), has a monograph in the European Pharmacopoeia (Ph. Eur. 10th Edition), European Medicines Agency (EMA), and European Scientific Cooperative on Phytotherapy (ESCOP). The medicinal plant material consists of whole or cut dried sage leaves containing not less than 12 or 10 mL/kg of essential oil, respectively. In addition to the essential oil containing monoterpenes and sesquiterpenes, the chemical composition includes diterpenoids, triterpenoids, flavonoids, hydroxycinnamic acid derivatives, and phenolic glycosides [110]. Traditionally, sage leaf can be used to relieve dyspeptic disorders such as heartburn and flatulence, reduce hyperhidrosis, relieve inflammation of the mouth and throat, and treat inflammation of the skin [111].

Medicinal products available on the Polish market are intended primarily for treating inflammations in the mouth and throat. These are concentrates for preparing rinse solutions (Dentosept, Salviasept, Tinctura Salviae, and Tymsal) and gels applied directly to the lesions within the oral cavity (Aperisan, Dentosept A, and Mucosit). In addition, sage leaf is available as a single herb for making infusions (Salviae folium), as well as in the form of herbal mixtures for making infusions for gargling (Septosan) and for use in mild inflammatory conditions of the female genitalia (Vagosan). The only medicinal product to be applied directly to the skin is sage ointment containing the ethanolic extract of *Salvia officinalis* leaves.

The traditional use of sage leaf has been partially supported by scientific research. Studies conducted on in vitro and in vivo models have confirmed its anti-inflammatory, antioxidant, and antimicrobial properties as well as the beneficial effects on wound healing [112].

Strong anti-inflammatory properties after topical application on a model of mouse ear edema induced by croton oil have been demonstrated for *n*-hexane and chloroform extracts from sage leaves. The component responsible for the anti-inflammatory activity was ursolic acid [113].

The essential oil of *S. officinalis* showed antifungal activity against clinical strains of dermatophytes isolated from skin and nails and also inhibited NO production by LPS-stimulated macrophages [114]. In in vitro studies, sage leaf oil showed stronger antibacterial properties against *S. aureus* and *P. aeruginosa* than penicillin and mupirocin. Studies on an in vivo model of infected wounds in BALB/c mice showed that the ointment containing 4% of the *S. officinalis* oil statistically reduced the number of bacteria compared to the control group and the mupirocin-treated group. Moreover, compared to the control group, there was a shortening of the inflammatory phase, acceleration of cell proliferation, increased collagen accumulation, revascularization, and re-epithelialization. The content of pro-inflammatory cytokines (IL-1β, IL-6, and TNF-α) decreased, and the content of growth factors (FGF-2 and VEGF) increased [115]. 

In another in vivo study in a Wistar rat model, the wound healing capacity of an ointment containing 3 and 5% hydroethanolic extract of sage leaves was tested. The time of wound closure and re-epithelialization was significantly shortened compared to the control group. Moreover, the formation of new blood vessels and the number of fibroblasts increased in the wound, improving the proliferation phase of healing [116]. The methanolic extract of *S. officinalis* leaves inhibited hyaluronidase, elastase, and collagenase activity in vitro. In vivo studies on Swiss albino mice inhibited the formation of wrinkles induced by UV exposure [117]. In the prospective randomized, double-blind placebo-controlled study, an area of the skin of healthy volunteers was irradiated to induce local erythema. An ointment containing 2% sage extract has been shown to reduce local inflammation and erythema to a similar extent as 1% hydrocortisone ointment [118]. Although sage leaves extracts and essential oil are widely used for skin problems, no reports on the interactions of this plant material with skin microbiome were published.

### 5.13. Sophora japonicum L. (Japanese Pagoda Tree Fruits)

The Japanese pagoda tree is also named sophora and belongs to the *Fabaceae* family. At least 153 constituents, including flavonoids, isoflavonoids, triterpenoids, alkaloids, mineral elements, and amino acids, were identified and isolated from *S. japonica*. The most important and abundant components of the dried flower buds and ripe fruits are rutin and sophoricoside [119]. These substances have been used to control the quality of medicinal products and determine the high medicinal value of this plant material. Based on the chemical composition, this plant material has anti-inflammatory, antibacterial, antiviral, and antioxidant effects and is traditionally used in treating wounds and trophic ulcers.

Sophora’s polysaccharides protect HaCaT keratinocytes from UVB irradiation-induced skin injuries and may involve the MAPK signaling pathway, which contributes to apoptotic cell death [120]. Sophoricoside ameliorates contact dermatitis due to the inhibition of the phosphorylation and degradation of IκB-α/β and the nuclear translocation of NF-κB p65 in B cells [121]. In turn, sophoricoside exhibited a potent inhibitory effect in the IL-5 bioassay in a dose-dependent manner [122]. This isoflavone glycoside inhibited the IL-6 bioactivity with an IC_50_ value of 6.1 µM. In contrast, it had no effects on IL-1β and TNF-α production and was established as a selective inhibitor of cyclooxygenase COX-2 activity [123].

In Ukraine, a tincture of Japanese sophora fruit (FITOPHARM, Kyiv, Ukraine) is very popular as an antiseptic and wound healing preparation and is used in purulent inflammatory processes (wounds and trophic ulcers).

### 5.14. Symphytum officinale L. (Comfrey Root)

*Symphytum officinale* is a plant belonging to the *Boraginaceae* family, rich in allantoin, phenolic acids (e.g., rosmarinic, *p*-hydroxybenzoic, caffeic, chlorogenic, and p-coumaric acids), pyrrolizidine alkaloids, triterpene saponins, tannins, amino acids, flavonoids, triterpenes, terpenoids, saponins, sterols, and mucopolysaccharides [124]. The comfrey plant material therapeutic properties include anti-inflammatory, analgesic, granulation-promoting, and anti-exudative effects [125,126].

The previous investigation determined the in vivo wound healing effects of *Symphytum officinale* L. leaves extract. The results showed that comfrey extract modulates the inflammatory process and stimulates collagen production [126]. Proliferative and antioxidant studies demonstrate a beneficial effect on human skin fibroblasts. It is non-toxic and simultaneously expresses the high ability to reduce ROS [125]. Several investigations with isolated compounds from comfrey were conducted, and it was established that crude comfrey polysaccharides possess the antioxidative activity and revealed that, by efficiency, it is superior to allantoin ointment in burn wound healing. Moreover, poly[3-(3,4-dihydroxyphenyl) glyceric acid from *S. asperum* and *S.caucasicum* roots inhibits the TNF-α production by human macrophages [127]. Furthermore, rosmarinic acid isolated from *Symphytum officinale* L. was shown to inhibit the formation of inflammation mediators of the arachidonic acid cascade in vitro [128]. Moreover, researchers conclude that comfrey root extract inhibits NF-κB by interfering with the activation pathway, at least in part at the level of IκB-α phosphorylation and possibly of IKK activation.

Mostly, medicines from comfrey root are presented in ointments (“Ointment Dr. Taissa with Comfrey”, Dr. Theiss Naturwaren GmbH, Homburg, Germany, and “Ointment with Comfrey”, DKP Pharmaceutical Factory, Zhytomyr, Ukraine) and used in the treatment of pain in the joints, back, lower back with radiculitis, osteochondrosis, arthritis, domestic injuries and bruises, sprains, dryness, and cracked skin. No research on the interaction of comfrey extracts with skin microbiome is available.

## 6. Conclusions

Plants have been used to prevent and treat skin diseases of various etiologies since ancient times. Due to their longstanding use, people have gained information regarding their effectiveness, active ingredients, as well as associated side effects. However, very often, it is not clear what mechanisms are responsible for the observed therapeutic effects. Moreover, recent advances in understanding the contribution of the skin microbiota in the maintenance of skin homeostasis can put new light on understanding the activity of topically applied plant-based products. Although the influence of various plants on skin-related ailments is well documented in vivo and in vitro, little is known about the interaction with the network of the skin microbial ecosystem, especially considering the prolonged treatment. It is also unclear whether skin microbiota can alter the chemical composition of herbal drugs applied directly on the skin surface. It was shown that some of the reported plant materials (e.g., sage leaves preparations) could have antimicrobial potential. However, available reports are strictly limited to investigating plant preparations influencing the growth of single strains of the chosen microorganisms. The analysis of the number of studies reported in Scopus between 2001 and 2021 using plant extract/material and skin microbiota as keywords showed that only 48 reports were found. Skin microbiota is a complex ecosystem that can certainly be modulated by plant extracts in many ways. Without solid basic studies involving human skin microbiota, the interaction of plant materials with the microbiome will remain unknown. Recently, there has been an arising interest in an investigation of the interaction between drugs and gut microbiota [129,130]. Based on the present review, it can be suspected that one of the major problems related to the lack of proper studies is the lack of well-described and reliable models that can be used for the investigation of the interactions between skin microbiome and plant extracts in vitro. More focus on this aspect of the problem is needed. For sure, future studies devoted to the investigation of skin microbiota with topically used plant materials or extracts are essential for the complex understating of the mechanism of action of those natural drugs in the prevention and treatment of various skin diseases.

## Figures and Tables

**Figure 1 ijms-23-09644-f001:**
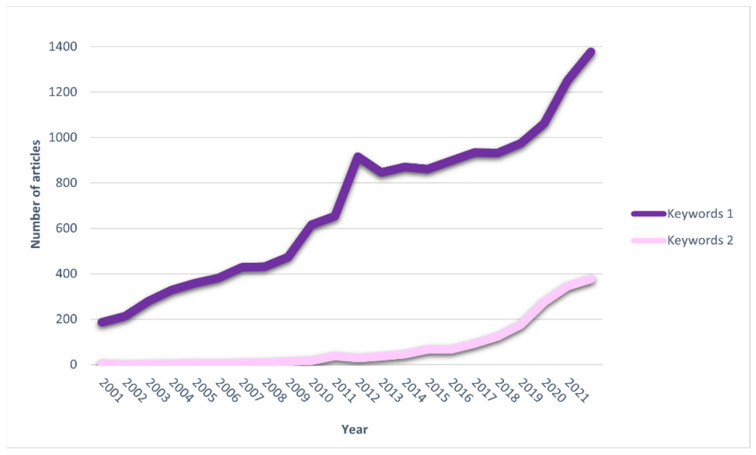
Graphical representation of the publication trend in the field of using plants related to their influence on the skin and microbiota. A graph was created based on the Scopus database, keywords 1 (plant material or plant extract and skin) and keywords 2 (plant material or plant extract and microbiota) were used.

**Table 1 ijms-23-09644-t001:** Traditionally used plant material in skin disorders in Poland and Ukraine.

Botanical Name	Common Name	Part Used	Application Properties [24,25,26]	Traditional Use/Therapeutic Area [24,25,26]
*Achillea millefolium* L.	yarrow	herb	antimicrobial, anti-inflammatory, antioxidant, antiproliferative, and cytotoxic	small superficial wounds
*Agrimonia eupatoria* L.	agrimony	herb	anti-inflammatory and antioxidant	minor inflammation and small, superficial wounds
*Allium cepa* L.	onion	bulbs	anti-inflammatory, antioxidant, and antimicrobial	insect bites treatment, wounds, minor burns, boils, warts, and treatment of bruises
*Aloe vera* (L.) Burm.f.	aloes	leaves	antiproliferative	bedsores
*Artemisia absinthium* L.	wormwood	herb	antiseptic and anti-inflammatory	boils, wounds, and bruises
*Arctium lappa* L.	burdock	root	antibacterial, antiviral, antioxidant, anti-inflammatory, antiallergic, antimutagenic, and antiproliferative	seborrheic skin conditions
*Arnica montana* L.	arnica	flowers	anti-inflammatory, antimicrobial activity, and antioxidant, immunotoxic, cytotoxic, and anti-platelet	inflammation
*Bidens tripartita* L.	three-lobe beggartick	herb	anti-inflammatory, bactericidal, and hemostatic	diathesis
*Calendula officinalis* L.	calendula	flowers	wound healing, antiviral, antimicrobial, anti-inflammatory, antioxidant, photoprotective, repellent, and anti-irritative	sunburn, minor wounds, and minor inflammations
*Chelidonium majus* L.	greater celandine	herb	antiseptic and fungicidal	warts, callus, corns, pimples, shingles, eczema, and skin tumors
*Equisetum arvense* L.	field horsetail	herb	antibacterial, antioxidant, anti-inflammatory, and wound healing	superficial wounds
*Hamamelis virginiana* L.	hamamelis	leaves, bark	antibacterial, anti-inflammatory, antiviral, and radical-scavenging	minor inflammation and dryness
*Hippophae rhamnoides* L.	sea-buckthorn	fruits	anti-inflammatory, bactericidal, analgesic, and epithelializing properties	rashes, eczema, burns, bedsores, frostbite, ulcers that do not heal well, and radiation skin diseases
*Hypericum perforatum* L.	st. John’s wort	herb	anti-inflammatory wound healing	skin disorders and minor wounds
*Linum usitatissimum* L.	linseed	seeds	antiproliferative	trophic ulcers, burns, and radiation damage
*Melissa officinalis* L.	melissa	leaves	anti-inflammatory, antiviral, antimicrobial, antioxidant, and anti-inflammatory	external remedies for herpes
*Matricaria chamomilla* L.	chamomile	flowers	anti-inflammatory, antimicrobial, and wound healing	irritation, minor inflammation, sunburn, superficial wounds, and furuncles
*Plantago lanceolata* L.	ribwort plantain	leaves, herb	anti-inflammatory, antibacterial, antiviral, antioxidant, and analgesic	Boils, edema, and insect bites
*Potentilla erecta* L.	tormentil	rhizomes	antibacterial, antioxidant, and antitumor	minor inflammations
*Quercus robur* L.	common oak	bark	astringent and anti-inflammatory	minor inflammation
*Salvia officinalis* L.	sage	leaves	antimicrobial, antioxidant, anti-inflammatory, cytoprotective, and wound healing	minor inflammations
*Sophora japonica* L.	Japanese pagoda tree	fruits	hemostatic, antiseptic, and wound healing	wounds, trophic ulcers, and seborrheic dermatitis
*Symphytum officinale* L.	comfrey	roots	anti-inflammatory, wound healing, and antibacterial	pain and inflammation
*Trigonella foenum-graecum* L.	fenugreek	seeds	anti-inflammatory and antiulcer	minor inflammations
*Urtica dioica* L.	nettle	leaves, herb	anti-inflammatory, analgesic, and local anesthetic	seborrheic skin conditions
*Viola tricolor* L.	wild pansy	herb	antibacterial, antioxidant, cytotoxic activity against cancer cells, anti-nociceptive, and anti-inflammatory	skin disorders, minor wounds, and mild seborrheic skin conditions

**Table 2 ijms-23-09644-t002:** Selected preparations marketed as non-prescription medicines or medical products containing plant material as an active ingredient in Poland (PL) and Ukraine (UA).

Plant Material	Country Code	Marketed Product	Active Ingredient(s)	Manufacturer	Pharmaceutical Form	Registered Indications
***Allium cepa* L. (onion bulbs)**	PL	Cepan	in 100 g of cream: 20.0 g ethanolic extract of *Allium cepa*, 5.0 g extract of *Matricaria chamomilla*, 5000.0 IU sodium heparin, and 1.0 g allantoin	Unia	cream	scars and keloids after burns and surgery; treatment of contractures; treatment of scarring of the eyelids; treatment of scars from boils, ulcers, and acne
	PL	Contractubex	in 100 g of gel: 10.0 g extract of *Allium cepa*, 5000.0 IU sodium heparin, and 1.0 g allantoin	Merz Pharma	gel	scars restricting movement, enlarged (hypertrophic, swollen, and keloid-shaped), unaesthetic postoperative scars, amputation scars, burn and accident scars, contractures of, e.g., fingers (Dupuytren’s contracture), tendon contractures caused by injuries, and scar shrinkage
***Arnica montana* L. (arnica flowers)**	PL	Arnithei	in 100 g of gel: 24.0 g arnica tincture	Dr. Theiss Naturwaren	gel	relieve bruises, sprains, and local muscle pain
	PL	Uzarin	in 100 g of gel: 1.0 g extract of *Arnica montana*, 1.0 g extract of *Calendula officinalis*, and 1.0 g aluminum acetate	Nes Pharma	gel	bruises, swellings, first-degree burns, and insect bites
***Calendula officinalis* L. (marigold flowers)**	UA	Marigold ointment	tincture of calendula flowers	Viola	ointment	burns, cuts, cracks in the skin, and purulent wounds
				DKP Pharmaceutical Factory		
				Lubnyfarm		
				Ternopharm		
				FITOPHARM		
	PL	Marigold ointment	ethanolic extract of *Calendula officinalis*	Elissa	ointment	mild inflammation of the skin, as an aid to the healing of minor wounds
	PL	Marigold ointment	extract of *Calendula officinalis* (extraction solvent: liquid paraffin)	Ziaja	ointment	symptomatic treatment of mild skin inflammations and as an auxiliary in the treatment of minor wounds (abrasions of the epidermis)
	UA	Marigold tincture	tincture of calendula flowers	Lubnypharm	tincture	drugs that promote wound healing
				FITOPHARM		
				Viola		
				Vishpa		
	PL	Marigold tincture	tincture of marigold flowers	FITOPHARM	tincture	symptomatic treatment of mild skin inflammations (such as sunburn) and as an adjunct in the treatment of minor skin wounds and mild inflammation of the mouth and throat
***Chelidonium majus* L. (greater celandine herb)**	UA	CHYSTOTIL	celandine extract	Khimpharmzavod Chervona Zirka	cream	bactericidal and wound healing
	UA	CHYSTOTIL	oil extracts of flowers, leaves, and roots of celandine	NATURE LIFE	ointment	analgetic, anti-inflammatory, and bactericidal
***Hippophae rhamnoides* L. (sea-buckthorn fruits)**	UA	Sea buckthorn ointment	sea buckthorn oil	Fitolic	ointment	healing (scarring) of wounds
	UA	Olasol spray	sea buckthorn oil—5.40 g; chloramphenicol—1.62 g; benzocaine—1.62 g; and boric acid—0.27 g	STOMA	spray	infected wounds, including long-term non-healing burns, trophic ulcers, and skin grafts
	UA	Mintalon	propolis, mummy, sea buckthorn oil, wheat germ oil, natural honey, geranium oil, terpene oil, vaseline oil, lecithin, pine resin, birch tar, camphor, and vitamin E	MINTA	balm	wounds, mechanical damage to the body surface, bruises; thermal damage (burns and frostbite); inflammation and purulent processes; bedsores; bumps; animal and insect bites; dryness and cracks of the skin; skin irritation; prevention of negative effects in frost, sun, wind; and moisturizing and normalizing skin nutrition
	UA	Reskinol	terpene oil, propolis, mummy, lecithin, sea buckthorn oil, pine resin, wheat germ oil, birch tar, natural honey, camphor, geranium oil, and vitamin E	Botany	balm	from pain and inflammation
***Linum usitatissimum* L. (linseed)**	PL	Linomag	virgin linseed oil	Ziołolek	Ointment, cream, or liquid	eczema, blemishes, and nappy rash: in states of excessive dryness of the skin; and to relieve the symptoms of psoriasis
	PL	Poldermin Hydro	*Linum usitatissimum* seed extract, *Avena sativa* extract, xylitol, and β-glucan	Polfa Tarchomin	Cream	intensively moisturizes the skin, softens it, and eliminates itching. In addition, it creates a protective film on the skin surface, accelerates the healing process of irritations, and reduces skin peeling
	PL	Aquastop Radioterapia	linseed oil and allantoin	Ziołolek	Cream	for skin care during and after radiotherapy
***Matricaria chamomilla* L. (chamomile flowers)**	PL	Chamomile ointment	ethanolic extract of *Matricaria chamomilla*	Elissa	ointment	skin inflammation
	PL	Kamagel	glycolic extract of *Matricaria chamomilla* and aluminum acetate	KRKA	Gel	inflammatory symptoms in various inflammations of the skin
	PL	Azulan	ethanolic extract of *Matricaria chamomilla*	Herbapol	Tinctura	in inflammation of the skin and mucous membranes, e.g., for rinsing in inflammations of the mouth and throat and inflammation of the gums
***Potentilla erecta* L. (tormentil roots)**	PL	Tormentil complex ointment (Tormentillae unguentum compositum)	in 100 g of ointment: 3.0 g liquid extract of the tormentil and 2.0 g ammonium bituminosulphonate 20.0 g zinc oxide	Unia	Ointment	treatment of minor skin lesions, such as skin abrasions and scrapes
				Amara		
				Ziaja		
				Prolab		
	PL	Tormentile forte	in 100 g of ointment: 3.0 g liquid extract of the tormentil, 2.0 g ammonium bituminosulphonate, 20.0 g zinc oxide, and 1.0 g borax (sodium tetraborate decahydrate)	Farmina	Ointment	skin lesions such as eczema lesions, first-degree burns, and mild acne vulgaris
	PL	Tormentiol	in 100 g of ointment: 2.0 g liquid extract of the tormentil, 2.0 g ammonium bituminosulphonate, 20.0 g zinc oxide, and 1.0 g borax (sodium tetraborate decahydrate)	Omega pharma	Ointment	minor skin damage such as abrasions and scratches. Incidentally, in purulent lesions and skin inflammations
***Quercus robur* L. (common oak bark)**	UA	BIOFLORIN	oak bark extract, coriander essential oil, and jojoba oil	Khimpharmzavod Chervona Zirka	cream	anti-inflammatory and wound healing
***Salvia officinalis* L. (sage leaves)**	UA	Tinctura Salviae	tincture of leaves salviae	DKP Pharmaceutical Factory	tincture	inflammation of the mucous membranes of the mouth, gums (stomatitis, gingivitis, periodontitis), pharynx, tonsils (pharyngitis, sore throat), upper respiratory tract, and infected wounds, cuts, and skin burns
	PL	Sage ointment	ethanolic extract of *Salvia officinalis*	Elissa	ointment	skin inflammation
***Sophora japonica* (Japanese pagoda tree fruits)**	UA	Tinctura sophorae japonicae	tinctura of Japanese sophora fruit	FITOPHARM	tinctura	antiseptics and disinfectants; purulent inflammatory processes (wounds and trophic ulcers)
***Symphytum officinale* L. (comfrey roots)**	UA	Ointment Dr. Taissa with Comrey	tincture of comfrey and tocopherol acetate	Dr. Theiss Naturwaren GmbH	ointment	the anti-inflammatory, analgesic effect, and promote the formation of calluses
	UA	Bainvel Comfrey Dr. Theiss	tincture of comfrey	Dr. Theiss Naturwaren GmbH	cream	degenerative-dystrophic and inflammatory diseases of the joints, as well as for recovery after sports and excessive or prolonged physical exertion
	UA	Ointment with Comfrey	comfrey root tincture and tocopherol acetate (vitamin E)	DKP Pharmaceutical Factory	ointment	pain in the joints, back, lower back with radiculitis, osteochondrosis, and arthritis. Sports, domestic injuries and bruises, sprains, and closed bone fractures; and dryness and cracked skin
**Another/Complex**	UA	Wundehil ointment	propolis, calendula, gooseberry foxglove, Japanese sophora fruit, and yarrow herb	AEM	ointment	remedies for wounds and ulcers
	UA	TRAUMEEL^®^ S	*Achillea millefolium* 0.09 g, *Aconitum napellus* D1 0.05 g, *Arnica montana* D3 1.5 g, *Atropa belladonna* D1 0.05 g, *Bellis perennis* 0.1 g, *Calendula officinalis* 0.45 g, *Echinacea* 0.15 g, *Echinacea purpurea* 0.15 g, *Hamamelis virginiana* 0.45 g, Hepar sulfuris D6 0.025 g, *Hypericum perforatum* D6 0.09 g, *Matricaria recutita* 0.15 g, Mercurius solubilis Hahnemanni D6 0.04 g, and *Symphytum officinale* D4 0.1 g	Biologische Heilmittel Heel GmbH	ointment	bedsores, burns, pityriasis, and trophic ulcers
	UA	Express BITE	*Chamomila recutita extract*, *Melaleuca alternifolia*, pantpenol, *Aloe arborescens extract*, *and Eugenia caryophyllus oil*	Georg BioSystems	cream	to eliminate skin itching
	UA	Express Burn	Chamomila extract Aloe extract Shea butter Tea tree oil Colloidal silver D-panthenol	Georg BioSystems	cream	treatment wounds and scars
	UA	UGRIN	*Millefolii herb*, *Menthae folia*, *Calendulae officinalis flores*, *Tanaceti flores*, *Lavandulae herba*, *Chelidoni herba*, and *Chamomillae recutitae flores*,	Khimpharmzavod Chervona Zirka	tincture	wound healing, anti-inflammatory, and antimicrobial action

## Data Availability

Not applicable.

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
