# Peer review of "Current Knowledge on Interactions of Plant Materials Traditionally Used in Skin Diseases in Poland and Ukraine with Human Skin Microbiota"

_ijms, 2022, doi:10.3390/ijms23179644_

Round 1
Reviewer 1 Report
After a careful proof-reading of the manuscript, this research article is recommended for publication in International Journal of Molecular Sciences, with some slight remarks and suggestions.
- References should be checked as some references are not properly formatted.
Author Response
Reviewer #1: After a careful proof-reading of the manuscript, this research article is recommended for publication in International Journal of Molecular Sciences, with some slight remarks and suggestions.
- References should be checked as some references are not properly formatted.
References were formatted accordingly.
Reviewer 2 Report
The present review is an interesting paper. I would recommend some suggestions:
The title shall be more attractive for readers.
Abstract needs to be rewritten, please highlight the novelty of this review and why now.
There are several long statements in the manuscript and references are missing and actually needed, e.g.:
“More and more drugs, dietary supplements, and cosmetic products appear on the 35 pharmaceutical and cosmetic market, which contain medicinal plant materials or sub- 36 stances of plant origin. In recent years, phytotherapeutic preparations are gaining greater 37 importance in solving many problems in dermatology and cosmetology”.
Conclusions and Future Perspectives - this section needs to be improved with more relevance related to this paper aim. It has a general character but should be more focused.
Add the list of acronyms or abbreviations.
The authors need to add the graphical representation figure and explanation of the publication trend (2000-2020) in the field of” Traditional use of topically applied plant materials in skin dis- 2 eases in Poland and Ukraine”.
Author Response
Reviewer #2: The present review is an interesting paper. I would recommend some suggestions:
- The title shall be more attractive for readers.
The title was changed.
- Abstract needs to be rewritten, please highlight the novelty of this review and why now.
The abstract was corrected.
- There are several long statements in the manuscript and references are missing and actually needed, e.g.:
“More and more drugs, dietary supplements, and cosmetic products appear on the 35 pharmaceutical and cosmetic market, which contain medicinal plant materials or sub- 36 stances of plant origin. In recent years, phytotherapeutic preparations are gaining greater 37 importance in solving many problems in dermatology and cosmetology”.
References were added wherever needed.
- Conclusions and Future Perspectives - this section needs to be improved with more relevance related to this paper aim. It has a general character but should be more focused.
Section “Conclusion” was corrected to point put most imported results of the current work.
- Add the list of acronyms or abbreviations.
The list of abbreviations was created and added to supplementary materials.
- The authors need to add the graphical representation figure and explanation of the publication trend (2000-2020) in the field of ”Traditional use of topically applied plant materials in skin dis- 2 eases in Poland and Ukraine”.
Figure 1 was created based on search in Scopus database and was summarized in the introduction section of the paper.
Reviewer 3 Report
The authors aimed to summarize current information on plant materials which are contained in pharmaceutical and cosmetic preparations available in Poland and Ukraine and marketed as non-prescription medicines or medical products. The paper presents the current knowledge regarding the traditionally used plant raw materials related to their influence on the skin and skin microbiota. Medicinal plants used in the treatment of wounds, burns, dermatitis, atopic dermatitis, eczema, and other inflammatory diseases of the skin have been included.
The study covers some issues that have been overlooked in other similar topics. The structure of the manuscript appears adequate and well divided in the sections. Moreover, the study is easy to follow, but some issues should be improved. Some of the comments that would improve the overall quality of the study are:
a. Authors must pay attention to the technical terms acronyms they used in the text.
b. English language needs to be revised.
c. Conclusion Section: This paragraph required a general revision to eliminate redundant sentences and to add some "take-home message".
Author Response
Reviewer #3:
The authors aimed to summarize current information on plant materials which are contained in pharmaceutical and cosmetic preparations available in Poland and Ukraine and marketed as non-prescription medicines or medical products. The paper presents the current knowledge regarding the traditionally used plant raw materials related to their influence on the skin and skin microbiota. Medicinal plants used in the treatment of wounds, burns, dermatitis, atopic dermatitis, eczema, and other inflammatory diseases of the skin have been included.
The study covers some issues that have been overlooked in other similar topics. The structure of the manuscript appears adequate and well divided in the sections. Moreover, the study is easy to follow, but some issues should be improved. Some of the comments that would improve the overall quality of the study are:
- Authors must pay attention to the technical terms acronyms they used in the text.
The acronyms were checked and a list of abbreviations was created and added as a supplementary material (Table S1).
- English language needs to be revised.
The paper was checked by a competent English speaker.
- Conclusion Section: This paragraph required a general revision to eliminate redundant sentences and to add some "take-home message".
Section “Conclusion” was corrected to point put most imported results of the current work.